# How Effective Is Water Technology as a Water Scarcity Remedy for the Economy in China?

**Na Qiao [1], Ai Yue [1,\*], Hongyu Guan [1], Lan Mu [2] and Yuxiu Ding [1]**

[1] Center for Experimental Economics in Education, Shaanxi Normal University, Xi'an 710119, China
[2] Northwest Institute of Historical Environment and Socio-Economic Development, Shaanxi Normal University, Xi'an 710119, China
\* Correspondence: joannaceee@163.com

**Abstract:** China's water shortage is severe and has become one of the factors hindering economic growth. It is widely accepted that changes in water technology play a profound role in sustainable development. However, because water technology is embedded in water resources, it is difficult to clarify the effects of water technologies as a means of alleviating water scarcity to achieve sustainable development. The level of water technology can be measured by total-factor water efficiency. This study analyzes panel data from 2002–2018 using the stochastic frontier model and addresses water inputs and water technology by introducing the concept of water productivity into economic growth framework. We found that, with the introduction of water technology, the importance of water resources for economic growth increases; water inputs increase by 1%, and GDP increases by 0.349%. The progress of 1% water technology can drive 0.142% of GDP growth and have great potential as a water scarcity remedy for the economy. Due to lower prices, the substitution rate of water technology to water is six times that of water-to-water technology, which is very detrimental to water technology progress. In the short term, water technology can be policy-driven in water-scarce developing economies, and price-induced in the long term.

**Keywords:** water technology; water; economic growth

## 1. Introduction

Water crises have been identified as one of the top five global risks to society in each of the last seven years according to the World Economic Forum [1]. Sixty-six percent of Chinese provinces had water resources of less than 2000 m$^3$ per capita in 2020, which means that they were in a state of moderate water shortage [2]. The shortage of water resources has restricted economic growth in China. A trade-off relationship exists between GDP output and water use in economic development. Increases in the GDP output of an economy entail requirements for more water resources. A series of studies have proven that water shortages restrict economic growth in China [3], and water is the bottleneck of economic growth in large cities such as Beijing and Tianjin [4]. Between 1981 and 2002, economic growth in China decreased by 0.1397% per year due to water shortage constraints. The value of drag was six times the figure in the United States [5]. Given the predictions of further change in the global climate, regional economies will suffer more severely from water scarcity constraints [6].

It Is widely accepted that water technological change plays a profound role in sustainable development. Neoclassical economists believe that technology is the reason why the modern economy has broken through the constraints of natural resources and achieved rapid growth [7,8]. Technological changes in the water resources sector can break through the constraints of the water crisis in terms of long-term economic growth [9]. Moreover, experiences in many regions of the world, such as Israel, have shown that the use of advanced water technologies to alleviate water shortages and promote the economy is quite

effective [10]. Since 2000, China has benefited from the advancement of water technology, and more than 8 billion m$^3$ of water has been saved every year. The amount of saved water was as high as 130.62 billion m$^3$ in 2017 [11].

However, Schultz claimed that technology is not a new factor and is embodied in various factors. Likewise, water technology is embedded in water resources. Therefore, the amount of water input is the result of water technology acting on water resources. As it is difficult to clarify the effects of water technologies on water, little is known about alleviating the economic constraints of water scarcity through water technology. Given its similarities to many other developing countries, China's situation might vary substantially from developed countries because of differences in advanced levels of water technology, political institutions, and the water market. There are more difficulties for backward water technology to break through constraints on the economy. Moreover, it is uncertain whether water technology is sufficient to meet the needs of sustainable development.

The effect of water technology as a water scarcity remedy for the economy is not well understood or studied to date. Most existing studies focus on the relationship between water and economic growth. First, a series of studies have proven that the water crisis restrains economic growth [12,13]. Other studies have explained the decoupling trend between water inputs and economic growth [14–16]. Studies have used a vector autoregressive model to evaluate the interaction between water resources and the economy [17,18]. There are only a few related studies that have addressed water technology in relation to the economy from a micro perspective [19]. Little is known about the impact of water and water technology on the economy given how water technology has become involved in production.

As water technology is embedded in the water resources used, the used water includes the impact of water technology, and the coefficients of water resources used are often underestimated. The contribution of water technology to economic growth is partially offset by the reduction in the amount of water inputs. A significant task for current research is to identify the basic mechanisms through which water technology impacts used water and determine the effectiveness of water technology as a water scarcity remedy for the economy. This paper attempts to address this issue.

First, we identify the mechanism between water technology change and water input by introducing the concept of water productivity. Analysis of the case of China indicates that water technology is not an independent production factor but dependent on water input. As such, the concept of water productivity can explain how water technology affects water inputs.

Second, we estimate the real contribution of water resources to economic growth by integrating water resources and water technologies into the framework of economic growth and explore the path for achieving sustainable economic growth in the context of water scarcity. The economic contribution of water inputs would be underestimated if water technologies were not considered. In this paper, the inclusion of water resources and water technologies in the analytical framework allows for a realistic estimation of not only the dependence between economic growth and water resources but also the extent to which water technologies support economic growth in China's unique environment.

Third, this paper provides important evidence for exploring an effective path to advance water technology in the developing world by focusing on China, where a unique empirical setting is created by the central government's use of high-powered political incentives to enforce the use of water technology.

It should be noted that the water shortage is not an absolute shortage, but rather a relative shortage of water use. Therefore, this paper focuses the study perspective on the water use.

## 2. Water Technology

### 2.1. The Concept of Water Technology

In this paper, water technology not only refers to technological innovation but also includes technology dissemination and adoption of innovation. O'Callaghan argued that the real issue of water technology is not so much the extent of research and development efforts but the rate of diffusion of technologies and adoption of innovation [20]. To better understand the concept of water technology, this paper uses the definition of Kumbhakar and Lovell [21], who separated total factor productivity into technical efficiency change, technical progress, scale efficiency change, etc. Technological advances are determined by technological innovation and the introduction of advanced technology, which drives economic growth by improving water efficiency and reducing water inputs. In China, water technology is very underdeveloped, and technological innovation is poor. Due to the lack of funds, talent, etc., water technology has progressed particularly slowly, despite technological innovation. Therefore, water technology progresses mainly through the introduction of foreign advances. As a spillover effect of technology, each province has a common rate of technological progress.

The diffusion of technologies and adoption of innovation can be described as the process of learning by doing. Arrow [22] observed that technical change in general can be ascribed to experience, and described technical change as a vast and prolonged process of learning. However, the diffusion of technologies and adoption of innovation can also be expressed in terms of technical efficiency. The reasons for low levels of technical efficiency are complex and inconclusive, with some scholars attributing it to institutions, infrastructure, and technological innovation [23], while others suggest that human capital, urbanization, and economic structure are the most significant factors in improving technical efficiency [24].

### 2.2. Identifying the Water Technologies Involved in the Production

The water crisis exists to the extent that economic growth has become increasingly dependent on water resources. Economic growth slows due to reduced water input. However, recent empirical data analyses have shown that economic growth is decoupled from water resources and that economic growth is not entirely dependent on water resources [25]. One possible reason that the empirical research comes to these conclusions is that the water crisis is not yet serious. However, this notion is inconsistent with the facts. Another reason is that the important factor, water technology, has been neglected. Few recent articles related to the impact of water on the economy have included water-saving technology, which builds on the concept of general technological progress and explains how water technology is involved in production.

However, water technology is an important driver of economic growth in a water scarcity scenario. Romer assumed that growth is driven by technological change as technology is nonrival and partially excludable [26]. Responding to water shortages, a variety of water-saving technologies will be adopted that arise from decisions made by utility-maximizing water users, thus water technological change arises. As it is nonrival, water technology can be accumulated without bound on a per capita basis. Hence, growth is driven fundamentally by the accumulation of water technologies.

Part of the water changes themselves imply changes in water technology and are the product of advances in water technology. Water input changes because of the substitution or supplementation of water technology with water inputs. As water inputs decline, the economy does not decline because the progress of water technology can substitute the water and increase or keep water productivity constant.

The concept of water productivity can describe the relationship between water and water technology. Lucas' [27] work takes effective labor, which includes both the number and skill level of workers, as a production factor in human capital theory. Similar to the concept of effective labor, the number of factors and the productive capacity are involved in production. We can define water productivity based on both numerical and technological

properties. Taking water-saving irrigation as an example, the water utilization rate of drip irrigation is 80%, that of flood irrigation is 40%, and the water irrigation demand is 80 m$^3$. Moreover, 100 m$^3$ of water is needed to meet irrigation demands through the adoption of drip irrigation technology, while 200 m$^3$ of water is needed to adopt flood irrigation systems without any water technology.

Based on the endogenous technological change in the DICE (ENTICE) model [28], which adds induced energy technological change to build an energy technology innovation model, water productivity can be described by the C-D function. First, the marginal output of water is positive, and there are diminishing returns from "learning by doing" over time. The partial derivatives of water resources and water technology are positive, but the second derivatives are negative. Romer used the Equations (1)–(4) to represent [29]:

$$\frac{\partial E}{\partial W} > 0, \frac{\partial^2 E}{\partial W^2} < 0 \tag{1}$$

$$\frac{\partial E}{\partial H} > 0, \frac{\partial^2 E}{\partial H^2} < 0 \tag{2}$$

Second, as water resources used (or water technologies) tend toward zero, marginal water productivity (or water technologies) tends toward infinity. Moreover, as water resources used (or water technologies) tend toward infinity, the marginal water productivity (or water technologies) tends toward zero. Both of the above cases meet the Inada conditions:

$$\lim_{W \to 0} \frac{\partial E}{\partial W} = \lim_{H \to 0} \frac{\partial E}{\partial H} = \infty \tag{3}$$

$$\lim_{W \to \infty} \frac{\partial E}{\partial W} = \lim_{H \to \infty} \frac{\partial E}{\partial H} = 0 \tag{4}$$

Given this, water productivity is

$$E_t\left(H_{E,t}\right) = A_w H_{E,t}^b W_{E,t}^\varphi \tag{5}$$

where $0 < b < 1$ and $0 < \varphi < 1$ are the elasticities of water technology and water inputs, respectively, $E$ is water productivity, $H$ is water technology, and $W$ is water input. This concept represents the change in water inputs via water technology and effectively explains the phenomenon that the decrease in water input caused by water technological progress does not reduce economic growth.

Water is an essential production factor for long-term economic growth. All forms of production, as well as people's lives, are inseparable from water. It is necessary to include water resources in the production function.

Therefore, imposing Equation (5) on the C–D production function, we obtain

$$Y = A_t K^\alpha L^\beta \left(H_{E,t}^b W_{E,t}^\varphi\right)^{1-\alpha-\beta} \tag{6}$$

where $Y$ is the output, $K$ is the capital stock, $L$ is the labor input, $W$ is the water input, $A$ is the technology in other sectors, and $H$ is the water technology.

It must be noted that the water price is not considered because it fluctuates only slightly and is strictly controlled in China. The central government's use of high-powered political enforcement to limit the amount of water consumption and set the water price creates a unique empirical setting in which water prices are very low and fluctuate very little. For example, the first-tier residential water price per cubic meter in Beijing, which is facing the worst water shortage with only 164.17 m$^3$ of water per capita, was only 5 yuan (0.76 dollars) from 2014 to 2021. Therefore, the inducing effect of price on water-saving technologies is relatively weak.

## 3. Data

### 3.1. Data of GDP, Capital, Labor, Water Input

To reduce the impact of missing variables and capture the dynamic information of each province, this paper adopts panel data that include 30 provinces (except Tibet) in China from 2002 to 2018. Tibet is not included because of the lack of the capital stock data. Since 2002, water technology has developed rapidly, so the data that this paper covers was collected from 2002 to 2018. The explained variable is GDP ($Y$), and the data are from the National Statistical Yearbook and are based on constant 2002 prices. The explanatory variables include capital input ($K$), labor input ($L$), water input ($W$) and water technology ($H$). The data of capital input ($K$) are from the National Statistical Yearbook and calculated by the perpetual inventory method with an economic depreciation rate of 10% using constant prices with 2002 as the base period. The labor input ($L$) data are from the National Statistical Yearbook. The water input ($W$) data are from the provincial Water Resources Bulletin. All the data are logarithmized to eliminate data fluctuation and heteroscedasticity.

The water technology variable is represented by total-factor water efficiency, which is calculated by data envelopment analysis (DEA) in this paper.

### 3.2. Water Technology in China

#### 3.2.1. Total-Factor Water Efficiency

Total-factor water efficiency is a composite indicator that reflects the integrated relationship between water inputs and economic outputs. Unlike single indicators, such as water consumption per 10,000 yuan of GDP, composite indicators are closer to reality, and the results of the analysis are more credible.

In previous studies, total factor productivity (TPF) was considered technical progress [30]. Various scholars have decomposed total factor productivity. Koopmans (1951) [31] provided the following definition of technical efficiency: a producer is technically efficient if, and only if, it is impossible to produce more of any output without producing less of some other output or using more of some input. Farrell [32] was the first to empirically measure productive efficiency and showed how to separate cost efficiency into its technical and allocative components. Total factor productivity was divided into technical efficiency change, technical progress, scale efficiency change, etc. (Kumbhakar and Lovell).

Based on the aforementioned description of water technology, technological progress can be measured by the distance shifted along the frontier, which is called "slack" and denotes the contribution of the shift in the production frontier to the change in productivity. Technical efficiency, which can be measured by the distance from an inefficient point to the projected point on the frontier and is called "radial adjustment", is a good description of the human capital, institutional, economic structure, and management factors in the "learning by doing" process.

Total-factor water efficiency established by Hu [33] with an index of the water adjustment target ratio from the production frontier was constructed using DEA. It is an effective ratio used to evaluate water efficiency among regions from the viewpoint of total factor production, which has been proven by empirical analysis.

Based on this framework, we can obtain the water efficiency value that represents the level of water technology. Given the assumption of constant returns to scale and keeping the output unchanged, the Equations (7)–(9) are all from the input-oriented Charnes, Cooper, and Rhodes (CCR) model [34]. Through mathematical programming to project the decision-making unit (DMU) onto the frontier, the envelopment of the $i$th DMU can be derived from the following linear programming problem:

$$max \sum_{r=1}^{q} u_r y_{rk} \tag{7}$$

$$\text{s.t.} \sum_{r=1}^{q} u_r y_{rj} - \sum_{i=1}^{m} v_i x_{ij} \leq 0 \tag{8}$$

$$\sum_{i=1}^{m} v_i x_{ik} \leq 1 \tag{9}$$

where $x$ values are inputs, $y$ is the output, $m$ is the category of input, $q$ is the $i$th category of output, $n$ is the number of DMUs, $v_i$ is the weight of input, and $u_i$ is the weight of output.

$$v_i \geq 0,\, u_i \geq 0,\, i = 1,2, \ldots, m;\, r = 1,2, \ldots, r;\, j = 1,2, \ldots, n$$

For the $i$th DMU, the distance from an inefficient point to the projected point on the frontier is called "radial adjustment", and the distance shifted along the frontier is called "slack". The summation of slack and radial adjustment for input is the total amount that needs to be adjusted to reach a "target" input while keeping the output unchanged, meaning that the amount needs to be reserved. Therefore, the total-factor water efficiency representing water technology, which is expressed by both Equation (10) and from Hu [32], is as follows:

$$WTE_{i,t} = \frac{TWI_{i,t}}{AWI_{i,t}} = \frac{AWI_{i,t} - LWI_{i,t}}{AWI_{i,t}} = 1 - \frac{LWI_{i,t}}{AWI_{i,t}} \tag{10}$$

where $WTE$ is the total-factor water efficiency, $AWI$ is the actual water input, $TWI$ is the water reduction target, $LWI$ is the radial adjustment of the water input, $i$ is the $i$th province, $t$ is the $t$th time, and $WTE$ falls between [0, 1]. In the above case, $LWI$ is higher, $WTE$ is lower, the amount of reserved water is increased, and the water technology is relatively backward. When $WTE = 1$, there is no wasted water, and water technology is in its most advanced stage. When $WTE < 1$, some water input can be saved.

3.2.2. Water Technology in China

The output index is *GDP*, and the input indexes include capital (*K*), labor (*L*), and water resources (*W*). Results by province are reported in Appendix A Table A1.

(1) The total WRT in China

The total value of WRT is 0.48 in China, which means that 52% of the water inputs were wasted in China during this period. If all provinces were at the forefront of water technology, then the amount of water that could have been saved would be approximately 313 billion m$^3$. Meanwhile, the change in the average level of total-factor water efficiency from 2002 to 2018 fluctuated around 0.5. From 2002 to 2007, total-factor water efficiency fluctuated upward; it decreased from 2007 to 2012, and steadily increased after 2012. In 2018, the WRT value was the highest at 0.507.

(2) Comparison of the average WRT between provinces

The average WRT of Yunnan, Shanghai, Liaoning, and Tianjin was close to 1, which means that the water efficiency was the highest and the water technology was the most advanced in these regions. The WRT value was the lowest in Xinjiang, at only 0.05. The WRT values were below 0.2 in Gansu, Guangxi, Jiangxi, and Ningxia. In 2018, the WRTs of Beijing, Shanghai, Tianjin, Liaoning, and Yunnan were the highest, while those of Xinjiang, Ningxia, and Qinghai were the lowest. It can be concluded that WRT is closely related to economic development, which is also related to the industrial structure and advanced water-saving equipment and facilities.

(3) Comparison of the WRT change between provinces

The WRT of most provinces increased overall, especially in Zhejiang, which increased from 0.54 (2002) to 0.79 (2018); in Fujian, which increased from 0.49 (2002) to 0.66 (2018); and in Shandong, which increased from 0.6 (2002) to 0.75 (2018). In contrast, Xinjiang's WRT dropped from 0.09 (2002) to 0.05 (2018).

## 4. Results

*4.1. Econometric Model*

A stochastic frontier model has advantages relative to a typical frontier model in identifying the unobserved heterogeneity across provinces and stochastic events. If the

differences across 30 individual provinces can be effectively identified, then the estimation effectiveness can be greatly improved over that of a model with a fixed front surface.

Equation (6) is expressed in logarithm form as Equation (11), which was modeled by Aigner [35].

$$\ln Y(t) = A(t) + \alpha \ln K_{it} + \beta \ln L_{it} + (1 - \alpha - \beta)b \ln H_{it} + (1 - \alpha - \beta)\phi \ln W_{it} + v_{it} - u_{it} \tag{11}$$

where the explained variable is GDP ($Y$), and the data are from the National Statistical Yearbook and are based on constant 2002 prices. The explanatory variables include capital input ($K$), labor input ($L$), water input ($W$), and water technology ($H$), where $i$ is the $i$th province; $t$ is the $t$th year; $\mu_{it}$ is the inefficiency term representing the distance from the $i$th province to the production frontier in the $t$th year, $u_{it} \sim N + (u, \sigma_u^2)$, $u_{it} \geq 0$; and $v_{it}$ is the random error, $v_{it} \sim N(0, \sigma_v^2)$. $\beta_0$, $\Phi$, $\alpha$, $\theta$, and $\lambda$ are the parameters to be estimated. Exogenous technological progress is characterized by the time trend term, $t$. The short definitions and summary statistics of the key variables are presented in Table 1.

**Table 1.** Variables description.

| Variables | Mean | STDev. | Minimum | Maximum |
|---|---|---|---|---|
| GDP (Hundred million RMB) | 11,224.31 | 10,870.61 | 330.17 | 66,257.88 |
| Capital (Hundred million RMB) | 33,232.12 | 37,278.82 | 684.6 | 192,708.4 |
| Labor (Ten thousand) | 2564.84 | 1698.05 | 282.4 | 6808 |
| Water (Hundred million m$^3$) | 195.42 | 138.06 | 19.96 | 592 |
| Total-factor water efficiency | 0.483 | 0.287 | 0.032 | 1.00 |

A correlation matrix is shown in Table 2. All inputs have positive correlation coefficients with the output, which explain that all inputs satisfy the isotonicity property with output. Only total-factor water efficiency has a negative correlation coefficient with water, which explains the substitution relationship between the two. The high correlation reveals that proportional relationships do exist between GDP and both capital and labor. While lower correlations explain that water input and water technologies do not perform the significant inputs to generate economic output.

**Table 2.** The correlation matrix for variables.

| Variables | GDP | Capital | Labor | Water | Total-Factor Water Efficiency |
|---|---|---|---|---|---|
| GDP | 1.000 | | | | |
| Capital | 0.870 | 1.000 | | | |
| Labor | 0.691 | 0.512 | 1.000 | | |
| Water | 0.461 | 0.324 | 0.537 | 1.000 | |
| Total-factor water efficiency | 0.206 | 0.108 | 0.093 | −0.376 | 1.000 |

First, the panel data were tested by the Hausmann test, and the null hypothesis was significantly rejected. A fixed effects model was used. Simultaneously, through regressing with a fixed effects model, the F test significantly rejected the null hypothesis that $u_i = 0$, which means that there were individual effects. The results estimated by ordinary least squares (OLS) regression are biased and inconsistent. Moreover, the coefficient of variation in the stochastic frontier model is Equation (12) [35]:

$$\rho = \frac{\sigma_u^2}{\sigma_u^2 + \sigma_v^2} \tag{12}$$

The estimate of $\rho$ is 0.994, which shows that the deviation was mainly determined by technological inefficiency $u_i$. Considering that technological efficiency may have a time trend, a time-varying model was adopted in the stochastic model framework in Equation (13) [35]:

$$u_{it} = e^{-\eta(t-T_i)} u_i \tag{13}$$

If $\eta = 0$, then technological efficiency did not change over time. The regression result was 0.0007, meaning that technological efficiency had nearly no time trend. Hence, we ignore the time trend effect of technological efficiency in Table 3.

**Table 3.** Econometric estimates of frontier economic effects of water technology.

| | Stochastic Frontier | | Mean Frontier | |
|---|---|---|---|---|
| | **(1)** | **(2)** | **(3)** | **(4)** |
| Capital | 0.323 *** (22.15) | 0.324 *** (21.16) | 0.321 *** (22.07) | 0.323 *** (22.13) |
| Labor | 0.060 * (1.75) | 0.053 * (1.47) | 0.031 (0.94) | 0.028 (0.80) |
| Water technology | 0.142 *** (2.47) | | 0.127 *** (7.13) | |
| Water | 0.349 *** (6.69) | 0.234 *** (6.87) | 0.332 *** (9.19) | 0.228 *** (−25.99) |
| time | 0.056 *** (23.76) | 0.058 *** (25.34) | 0.057 *** (26.67) | 0.058 *** (25.89) |
| constant | −109.116 *** (−23.78) | −111.219 *** (−25.31) | −111.340 *** (−23.78) | −113.006 *** (−25.99) |
| b | 0.230 | | 0.196 | |
| φ | 0.566 | | 0.512 | |

Note: Z and T statistics are in parentheses for each explanatory variable. *, ***, indicate 10% and 1%, respectively. The parentheses denote the value of prob > z or prob > t.

### 4.2. Results of the Impacts of Water Technology on the Economy

Table 3 shows the estimated coefficients of the stochastic frontier model with water technology (1) and without water technology (2) and the fixed effects model with water technology (3) and without water technology (4). The estimated coefficients of the stochastic frontier model are more significant than those of the fixed effects model because the effects of both unobserved heterogeneity stochastic events and other types of stochastic noise are considered. In the stochastic frontier, the estimated coefficients *K*, *H*, and *A* are significant at the 1% level, and *L* is significant at the 10% level. In the mean frontier, except for the estimated coefficient of *L*, the others are all significant at the 1% level.

(1) Impacts of water resources on economic growth

There is strong evidence that with the introduction of water technology, the importance of water resources for economic growth increases. On the stochastic frontier, with water technology, the estimated coefficient of water was 0.349 and was significant at the 1% level, which was higher than the capital coefficient (0.323). This means that water input increased by 1% and GDP increased by 0.349%. The contribution of water to economic growth was underestimated, as the risk of water crises was previously underestimated. While water technologies were not endogenous to water resources, the portion of the water resource that was reduced due to water technology could not be identified. Therefore, the water coefficient was lower because a portion of water's contribution to the economy was

offset by water technology. The coefficients without water technologies were 0.234 (on the stochastic frontier) and 0.228 (on the mean frontier). After adding water technology into the model, both water coefficients increased by approximately 0.1 and were higher than the capital coefficient. This illustrates the important challenges facing China's economic growth, with the declining role of capital as a driver. The risk to economic growth from the water crisis will further increase.

(2)    Economic impacts of water technology

The estimated coefficient of water technology was 0.142 (on the stochastic frontier), which was significantly different from 0 at the 1% level. This finding implies that the total-factor water efficiency increased by 1%, which means that 1% of water was saved and the GDP rose by 0.142%. This indicates that, due to advances in water technical progress and improvements in technical efficiency, water technologies can simultaneously replace 1% of water and drive 0.142% of GDP growth. This finding provides empirical evidence that water technology is indeed an important way to alleviate the effects of water resource constraints on economic growth and achieve win–win water savings and economic growth. It was previously mentioned that 52% of water resources can be saved by using the most advanced level of technology available in China. Therefore, water technology has great potential as a water scarcity remedy for the economy and is an important tool to achieve sustainable economic development in a water crisis situation. In addition, given the weak innovation in water technology in China, this also provides a good experience for developing countries in that, in addition to market mechanisms, the introduction of water technology and the top-down "learning-by-doing" process is another way to sustain development in a policy environment with high intensity restrictions on water quantity and water efficiency.

(3)    The relationship between water resources and water technology

To date, water technology is not often used as a substitute for water resources. The water productivity values are mainly due to the quantity of water input. According to Table 3 and Equation (5), as water input increased by 1%, water productivity rose by 0.566%. When water technology increased by 1%, water productivity rose by 0.233%. This finding shows the limited substitution of water technology for water resources. It took 2.46% of WRT to substitute for 1% of water inputs. Water inputs were increased by 0.406% to achieve a substitute of 1% of WRT. The substitution rate of water technology to water is six times that of water inputs to water technology. At this stage, China's water use pattern is still inefficient and the effects of water technology are limited.

*4.3. Robustness Test*

To verify the robustness of the regression results, this paper used the statistical data of the water-saving irrigation area to replace the total-factor water efficiency to prevent estimation bias and improve estimation effectiveness given calculation deviations. The stochastic frontier model and fixed effects model are still used for estimation.

The data on water-saving irrigation areas are used to replace the data on total-factor water efficiency because they have some interesting characteristics. First, water-saving irrigation accounts for the largest proportion of investment in water technologies. China's agricultural water consumption has long accounted for approximately 75% of total water consumption, and irrigation water can account for more than 90% of total agricultural water consumption. Promoting modern agricultural irrigation technologies has been identified as an important measure for combatting water scarcity [36]. Therefore, the major water-saving task is saving agricultural irrigation water. Second, the consideration of water-saving irrigation areas can prevent inaccurate estimates of depreciation and are a better measure of the level of water technology than investments in water departments are. Third, water-saving irrigation areas are not affected by price fluctuations, and there is no error between the nominal and actual prices. Finally, water-saving irrigation areas need to meet the needs of water technology, which is mainly driven by the total capital investment in the previous

period and the new investment in the current period. The judgment standard of the water-saving irrigation area is to reach a certain level of technological equipment, which represents the degree of promotion of water technology and its advanced level. Overall, the data from the "China Water Conservancy Yearbook (2002–2018)" of agricultural water-saving irrigated areas (*IR*) represent the water-saving technology accumulated by investment in innovation.

First, the P value of the Hausmann panel data test was 0.000, which means that the null hypothesis was strongly rejected. Therefore, in the OLS regression, the fixed effects model was used instead of the random effects model. The regression estimation results of the fixed effects model are shown in Table 4. Except for the coefficient of labor, the difference is small between the values in Table 3. The coefficient of labor in the OLS regression was 0.043 (7), while it was 0.067 (5) in the frontier stochastic model. Moreover, the estimated coefficient of water and water technology was smaller, which could be because the water-saving irrigation data involved only the agricultural industry, and thus its explanatory power was relatively weak. In this study, the results from using total-factor water efficiency in the regression were robust.

**Table 4.** Econometric estimates of water-saving irrigation.

| | Stochastic Frontier | | Mean Frontier | |
|---|---|---|---|---|
| | **(5)** | **(6)** | **(7)** | **(8)** |
| Capital | 0.346 *** (22.50) | 0.352 *** (23.11) | 0.345 *** (22.37) | 0.352 *** (23.01) |
| Labor | 0.067 ** (1.95) | 0.052 (1.52) | 0.043 (1.24) | 0.028 (0.82) |
| Water technology | 0.036 ** (2.47) | | 0.033 ** (2.27) | |
| Water | 0.217 *** (6.69) | 0.218 *** (6.67) | 0.211 *** (6.37) | 0.211 *** (6.34) |
| time | 0.053 *** (23.76) | 0.0541 *** (24.08) | 0.054 *** (24.20) | 0.055 *** (24.57) |
| constant | −102.933 *** (−23.78) | −104.044 *** (−24.00) | −104.893 *** (−24.36) | −105.905 *** (−24.62) |

Note: Z and T statistics are in parentheses for each explanatory variable. **, ***, indicate 5%, and 1%, respectively. The parentheses denote the value of prob > z or prob > t.

## 5. Discussion

(1) In China, the process of water technological progress is the introduction of new technologies and the process of "learning by doing". In particular, "learning by doing" is the driving force behind water technological progress. Aghion and Howitt [37] divided technological innovation into basic innovation (R&D) and secondary innovation (learning by doing). Romer [38] unified the concept of technological progress embodied in the improvement in capital quality with the process of capital accumulation and maintained that there is a fixed proportion of new capital and new knowledge. Due to the lack of core independent intellectual property rights and insufficient capital investment, water technology is very underdeveloped. Taking innovation investment as an example, since 2001, the Ministry of Water Resources has invested approximately 34 million yuan (5.32 million dollars) to promote technological innovation, which accounts for only 0.5% of the total annual investment in water conservancy construction. As a developing country, China can achieve higher industrialization and advanced technology and equipment based on those of developed countries [39]. Due to the implementation of policies of "the most stringent water resource system", water consumption has been strictly limited. Rational economic enterprises and humans are inclined to choose water-saving production methods or lifestyles. Material capital,

which is both abundant and inexpensive, is an effective carrier of new production knowledge and technology. Introducing foreign advanced technology is the fastest and most cost-effective way to improve water technology.

(2) Strong policies, especially goal-oriented policies, have accelerated the process of water technologies. The change in water technology was consistent with the changes in China's water-saving policies. The WRT value increased significantly in 2003, 2007, and 2012. In particular, from 2012 to 2018, it grew continuously. In China, a unique empirical setting is created by the central government's use of high-powered political incentives to force local governments to adopt a target-based water-saving system, such as limiting water consumption and improving water efficiency. In 2002, the construction of a national water-saving society began, denoted as the "Water Law"; pilot tests were conducted and an assessment measure was established. Therefore, the WRT increased significantly in 2003. In 2007, further policies regarding the building of a water-saving society were implemented. The national water consumption per 10,000 yuan of GDP was shown to decrease by more than 20%. In 2012, "Strict Water Resources Management System" policies began to be implemented, and "three red lines" were formulated: by 2030, the total water consumption of the country is to be controlled within 700 billion m$^3$, the water efficiency will approach the world's advanced level, and the total amount of major pollutants in rivers and lakes will be controlled. These objectives provide important empirical evidence that, in addition to market mechanisms, quantified goals, and bureaucratic incentives, are effective paths to saving water and advancing water technology in the developing world. Since the amount of water consumption and water efficiency are important for political promotion, local government officials have strong incentives to regulate water use. When the central government wants to mobilize local governments for the purpose of decentralized policy implementation, it often adopts a target-based incentive scheme in which political rewards are promised but are contingent on meeting certain performance criteria. The dissemination and adoption of new water technology is especially slow. The rate of innovation and technology uptake is often reported as being slower in the water sector than in other sectors due to the conservative nature of the industry [40,41]. O'Callaghan found that the adoption of a type of water technology in the study region took 12–14 years to move through the innovator and early adopter stages of the market and reach the early and late majority, and regulation and policy played a key role in encouraging the market to adopt these solutions [42]. However, strong goal-oriented policies will accelerate the progress of water technology.

(3) As the water shortages resulted in increasing constraints on economic growth, the estimated coefficients of water input should increase over time. The estimated coefficient of water input was 0.0298 (1978–2013) [43], and the province water coefficients ranged from −0.1 to 0.25 (1998–2013) [44]. In this paper, the water coefficient was 0.234 without water technology (2002–2018) and 0.349 with water technology. The GDP (constant price in 2000) associated with one cubic meter was 20.24 yuan (3.04 dollars) in 2000; in 2018, it increased to 4,608,333 yuan (692,632.45 dollars). As a result of strict water policies, water is allocated to sectors with higher GDP, such as the industrial sector, while water in the agricultural sector is withdrawn. Moreover, industries with high water consumption have had to constantly upgrade their water-saving technologies and improve their water economic efficiency. Therefore, the contribution of water resources to economic growth was more obvious than it had been before. Yang Yaowu and Zhang Ping [45] maintain that the contribution of the natural environment and resources to economic growth first decreased and then increased over time, showing a U-shaped curve. The contribution of water to the economy was found to be consistent with that of other resources. Therefore, the water elasticity coefficient of economic growth was low at first and increased over time. The development of water technology substituted for the contribution to the economy and lessened the dependent relationship between water and the economy.

(4)    The substitution rate of water technology to water was six times that of water inputs to water technology because the cost of water was much lower than that of water technology. The investment in drip irrigation projects was estimated to be approximately 1000 yuan (156.49 dollars) per acre. In addition, the average annual investment in other consumable products was approximately 400 yuan (62.60 dollars), which was too high for farmers to afford [46]. Moreover, the water price of agriculture was very low. The water amount per acre of rice was 180–320 m$^3$, and that of vegetables was 220–550 m$^3$. Taking Shaanxi as an example, the agricultural water price was 0.8 yuan (0.12 dollars)/m$^3$, the total annual rice water fee was approximately 144–256 yuan (21.64–38.48 dollars), and the vegetable water fee was 176–440 yuan (26.45–66.13 dollars). If the dropper technique was adopted, which could save 50% of water, 72–128 yuan (10.82–19.24 dollars) could be saved in rice cultivation, and 88–220 yuan (13.23–33.07 dollars) could be saved in vegetable cultivation, but these savings could not compensate for consumable product fees. Therefore, the adoption of water technology costs was significantly higher than the water input because of the low water price in China. Without strong government promotion, it is difficult to improve water technology through the inductive effect of water prices. Technology depends on policy, as has been proven in China, which is an important aspect of induced and biased technological progress.

## 6. Conclusions

Like many other water-scarce developing economies, China's water shortage is severe and has become one of the key factors hindering its economic growth. It is widely accepted that water technological change plays a profound role in sustainable development. However, water technology is embedded in water resources, and little is known about alleviating the economic constraints of water scarcity through water technology. By identifying the water technologies involved in production and introducing the concept of water productivity, this article attempts to clarify the mechanisms and impact of water technology as a water scarcity remedy for sustainable economic growth.

Exploiting the total-factor water efficiency, we estimate that, although the overall level of technology is advancing, 52% of water inputs can be saved at the current level of water technology. If all provinces were at the forefront of water technology, then the amount of water that could be saved would have been approximately 313 billion m$^3$ in 2018 alone. The water technology of Yunnan, Shanghai, Liaoning, and Tianjin was the most advanced in China.

We found that the importance of water resources for economic growth has increased with the introduction of water technology. When water inputs increase by 1%, GDP increases by 0.349%, which is higher than the capital contribution (0.323%). The contribution of water resources and the risk of water crises were underestimated.

This article quantifies the effects of water technology on economic growth by reducing water input. We found that water technologies could replace 1% of water and drive 0.142% of GDP growth during our study period (2002–2018). Therefore, because 52% of water resources can be saved, water technology has great potential as a water scarcity remedy for the economy and is an important tool to achieve sustainable economic development in a water crisis situation. This also provides a good experience for developing countries in that the water technology introduction and the top-down "learning-by-doing" process is another way to sustain development in a policy environment with high intensity restrictions on water quantity and water efficiency.

We also found that, as a result of low water prices, the cost of using water is much lower than the cost of adopting water technology. Thus, water users lack incentive to adopt water technology. Water technology has a limited role in substituting water resources. The substitution rate of water technology to water is six times that of water inputs to water technology. This is the key reason for the slow progress of water technology.

Driving innovation, application, and diffusion of water technology has great potential as a water scarcity remedy for the economy in the future. China provides a successful experience to many other water-scarce developing economies, which shows that countries lagging behind in water technology can accelerate the diffusion and adoption of introduced water technologies through policy instruments in a relatively short period of time. In the long term, by simultaneously increasing water prices and subsidizing water technologies, the adoption of water technologies will become economically effective. At this point, water technologies can be consistently promoted and water shortages can be effectively mitigated.

**Author Contributions:** Conceptualization, N.Q. and A.Y.; methodology, N.Q.; software, H.G.; writing—original draft preparation, N.Q.; writing—review and editing, A.Y., L.M. and Y.D.; supervision, A.Y.; project administration, L.M. All authors have read and agreed to the published version of the manuscript.

**Funding:** This research received no external funding.

**Data Availability Statement:** Data of labor are from the National Statistical Yearbook (http://www.stats.gov.cn/tjsj/) (accessed on 20 January 2022); The GDP are from the National Statistical Yearbook and are calculated based on constant 2002 prices. The data of capital input are from the National Statistical Yearbook and calculated by the perpetual inventory method with an economic depreciation rate of 10%, using constant prices with 2002. The water input data are from the provincial Water Resources Bulletin (http://www.mwr.gov.cn/) (accessed on 23 January 2022).

**Conflicts of Interest:** The authors declare no conflict of interest.

## Appendix A

**Table A1.** Total-factor water efficiency of province.

| Province | 2002 | 2006 | 2012 | 2016 | 2017 | 2018 |
|---|---|---|---|---|---|---|
| Beijing | 1.00 | 0.99 | 0.87 | 0.80 | 0.81 | 0.82 |
| Tianjin | 1.0 | 1.0 | 1.0 | 1.0 | 1.0 | 1.0 |
| Hebei | 0.48 | 0.54 | 0.52 | 0.55 | 0.55 | 0.56 |
| Shanxi | 0.67 | 0.69 | 0.55 | 0.52 | 0.54 | 0.55 |
| Inner Mongolia | 0.16 | 0.15 | 0.23 | 0.24 | 0.24 | 0.24 |
| Liaoning | 1.00 | 1.00 | 1.00 | 1.00 | 1.00 | 1.00 |
| Jilin | 0.38 | 0.42 | 0.27 | 0.29 | 0.31 | 0.32 |
| Heilongjiang | 0.27 | 0.28 | 0.21 | 0.22 | 0.21 | 0.22 |
| Shanghai | 1.00 | 1.00 | 1.00 | 1.00 | 1.00 | 1.00 |
| Jiangsu | 0.29 | 0.26 | 0.24 | 0.25 | 0.25 | 0.26 |
| Zhejiang | 0.54 | 0.57 | 0.63 | 0.75 | 0.77 | 0.79 |
| Anhui | 0.57 | 0.59 | 0.54 | 0.57 | 0.58 | 0.59 |
| Fujian | 0.49 | 0.54 | 0.53 | 0.64 | 0.62 | 0.66 |
| Jiangxi | 0.18 | 0.18 | 0.15 | 0.17 | 0.17 | 0.18 |
| Shandong | 0.60 | 0.65 | 0.66 | 0.71 | 0.74 | 0.75 |
| Henan | 0.50 | 0.52 | 0.44 | 0.47 | 0.47 | 0.48 |
| Hubei | 0.43 | 0.38 | 0.34 | 0.39 | 0.37 | 0.37 |
| Hunan | 0.32 | 0.35 | 0.34 | 0.36 | 0.36 | 0.35 |
| Guangdong | 0.40 | 0.49 | 0.49 | 0.54 | 0.54 | 0.57 |
| Guangxi | 0.18 | 0.18 | 0.14 | 0.15 | 0.16 | 0.23 |
| Hainan | 0.28 | 0.31 | 0.34 | 0.37 | 0.36 | 0.37 |
| Chongqing | 0.71 | 0.64 | 0.70 | 0.89 | 0.89 | 0.89 |
| Sichuan | 0.51 | 0.57 | 0.63 | 0.65 | 0.63 | 0.65 |
| Guizhou | 0.30 | 0.30 | 0.33 | 0.35 | 0.34 | 0.34 |
| Yunnan | 1.00 | 1.00 | 1.00 | 1.00 | 1.00 | 1.00 |
| Shaanxi | 0.42 | 0.41 | 0.40 | 0.40 | 0.39 | 0.40 |
| Gansu | 0.11 | 0.19 | 0.18 | 0.19 | 0.19 | 0.20 |
| Qinghai | 0.03 | 0.23 | 0.27 | 0.27 | 0.27 | 0.27 |
| Ningxia | 0.03 | 0.09 | 0.10 | 0.10 | 0.10 | 0.10 |
| Xinjiang | 0.09 | 0.04 | 0.04 | 0.05 | 0.05 | 0.0 |

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
