# Peer review of "How Effective Is Water Technology as a Water Scarcity Remedy for the Economy in China?"

_water, doi:10.3390/w14193056_

Round 1

Reviewer 1 Report

The authors investigated the effects of water technology on economic growth as well as the relationship with water productivity. While this subject is an important and critical issues to be addressed globally, this article may not be acceptable to be published with the current way of information organisation.

 The article needs major proofreading and paraphrase as there are many confusing sentences such as:

 “Water technologies can replace 1% of water… “-  please rephrase it for better clarity on how water technologies can replace 1% of water.

 “Between 1981 34 and 2002, economic growth in China decreased by 0.1397% per year due to water shortage constraints, which was 6 times the figure in the United States”. The 6 times is referring to economic growth or water shortage?

 The framework and methodology for the analysis of water technology & water productivity and how it is aligned to the objectives of this research is not clear. Hence, it is unclear how the results were obtained and why it is significant.

 The abstract and conclusion end with the message of “Since the cost of water is much lower than the cost of adopting water technology, water technology is little used as a substitute for water resources.” This scenario is evidently known based on previous publications. So, what is the recommendation/opportunity or way forward that the authors would like to suggest to address the water scarcity while facing this constraint?

Reviewer 2 Report

The study analyzes panel data from 2002-2018 using the stochastic frontier model and addresses water inputs and water technology by introducing the concept of water productivity into the economic growth framework. We find that with the introduction of water technology, the importance of water resources for economic growth increases, water inputs increase by 1%, and GDP increases by 0.349%. Comments: Add references to all of the equation, in the case You are not the Author of the equations. Line 274: Please, delete the following text: This section may be divided by subheadings. It should provide a concise and precise description of the experimental results, their interpretation, as well as the experimental conclusions that can be drawn. What lessons should water suppliers draw from this analysis? Are there concrete steps that can be recommended and how generalizable are the findings? Can they be applied to other areas?  How dependent are they to specific characteristics of the area under examination? 

Round 2

Reviewer 1 Report

-